# Impact of Active Video Games on Body Mass Index in Children and Adolescents: Systematic Review and Meta-Analysis Evaluating the Quality of Primary Studies

**DOI:** 10.3390/ijerph16132424

**Published:** 2019-07-08

**Authors:** Carlos Hernández-Jiménez, Raquel Sarabia, María Paz-Zulueta, Paula Paras-Bravo, Amada Pellico, Laura Ruiz Azcona, Cristina Blanco, María Madrazo, María Jesus Agudo, Carmen Sarabia, Miguel Santibáñez

**Affiliations:** 1Gerencia de Atención Primaria, Servicio Cántabro de Salud, 39011 Santander, Spain; 2Faculty of Nursing, University of Cantabria, Avda Valdecilla s/n., 39008 Santander, Spain; 3IDIVAL, Grupo de investigación en Enfermería, 39008 Santander, Spain; 4IDIVAL, GI Derecho Sanitario y Bioética, GRIDES, 39008 Santander, Spain; 5Care Continuity Coordinator, Área VI SESPA, Urbanización Castañeda s/n., 33540 Arriondas, Principado de Asturias, Spain; 6Hospital Universitario Marqués de Valdecilla, Servicio Cántabro de Salud, 39008 Santander, Spain

**Keywords:** active video games, body mass index, children, meta-analysis

## Abstract

Objective: To study the impact of active video games on Body Mass Index (BMI) in children and adolescents. Design and Methods: A systematic review and meta-analysis. Data were pooled in meta-analysis using the method of random effects or fixed effects, as appropriate, after examination of statistical heterogeneity. Data sources and eligibility criteria for selecting studies. A comprehensive literature research was conducted in Medline (PubMed), ISI web of Knowledge, and SCOPUS up to April 2018, in relation to clinical trials (both controlled and non-controlled) in children and adolescents, whose intervention was based on active video games. Results: The overall intragroup effect of the intervention based on active video games was in favor of the intervention, reaching statistical significance using the fixed effects model: (standardized mean difference (SMD) = −0.138; 95% CI (−0.237 to −0.038), *p* = 0.007 and was of borderline statistical significance in the random effects model: SMD= −0.191; 95% CI (−0.386 to 0.003), *p* = 0.053. The individual results of the determinations of the 15 included studies for this analysis showed a high heterogeneity among them (I^2^ = 82.91%). When the intervention was applied to children and adolescents with greater than or equal to 85 (overweight or obese) BMI percentile showed a greater effect in favor of the active video games: SMD= −0.483, *p* = 0.012. The overall intra-group effect in the control group was close to zero (SMD = 0.087). With respect to the non-standardized mean difference (MD) between groups, it was also in favor of active video games for both BMI (Kg/m^2^): DM = −0.317, 95% CI (−0.442 to −0.193), *p* = < 0.001 and BMI z-score: DM = −0.077, 95% CI (−0.139 to −0.016), *p* = 0.013. Conclusions: Our meta-analysis show a statistically significant effect in favor of using active video games on BMI in children and adolescents. The clinical relevance of this positive effect must be evaluated.

## 1. Introduction

Overweight and obesity are currently a serious public health problem. Its increasing prevalence also affects children and adolescents, with diet and physical activity being the most important modifiable factors for its prevention [1,2]. The idea of video games associated with sedentary lifestyles has undergone a radical change with the arrival of a new generation of video games, active video games, which involve physical activity since they allow players to interact physically, through their body movements (arms, legs, or the whole body), with the virtual reality that appears on screen through different devices.

Individual primary studies and subsequent meta-analyses have focused on the effects of active video games on the promotion of physical activity [3,4,5,6,7,8], energy expenditure [3,4,5,6,7,8,9,10,11,12,13,14,15,16,17,18,19,20,21,22,23,24,25,26], oxygen volume consumption [9,12,15,16,27,28,29], and heart rate [9,14,15,17,18,27,28,29,30,31,32]. These studies have shown an increase in heart rate and an increase in oxygen consumption. This energy consumption is statistically significantly higher compared to non-active video games and sedentary activities, despite the variability of the types of active video games, as well as the time spent playing them [5,7,19,22,23,24,25,26,33].

Regarding the effect of active video games on body mass index (BMI), various primary studies [21,31,33,34,35,36,37,38,39,40,41] have analyzed their effect using BMI as one of their dependent variables. However, no meta-analyses have been found focused on the BMI variable in children and adolescents.

Studies regarding the impact of active video games on BMI suggest that active video games can be a very good opportunity for reducing BMI, and with it, the prevalence of overweight and obesity, especially in children and adolescents. However, the heterogeneity in the design of the studies identified, in the study populations, in the measurement of the results, in the different types of active video games used and in the design of the control group, makes it difficult to interpret the different studies published in this respect. On the other hand, the varying quality and insufficient internal validity due to methodological weaknesses in some studies may also explain the differences in published results. At the same time, the sample size of most of the works reviewed was low, impeding the achievement of statistically significant results. Therefore, a meta-analysis synthesizing the quantitative findings in relation to BMI, and incorporating a subgroup analysis based on the quality of the identified studies and other methodological characteristics would be very useful.

## 2. Materials and Methods

### 2.1. Studies, Participants, and Interventions

A comprehensive literature research was conducted in relation to clinical trials (both controlled and uncontrolled) in children and adolescents, written in English or Spanish, whose intervention was based on active video games.

### 2.2. Outcome Measures

We included all studies that reported at least one determination of the BMI (Kg/m^2^, Z score or percentile) both pre- and post-intervention, in order to assess the intra-group change after the intervention with respect to the basal BMI.

Controlled studies reporting differences between groups in BMI were also included in order to assess the BMI between-group difference.

### 2.3. Search Strategy for Identification of Studies, Study Selection, and Data Abstraction

International bibliographic databases consulted were Medline (PubMed), ISI web of Knowledge, and SCOPUS. An electronic literature search was performed to identify all relevant studies (published, unpublished, either in press or in progress) up to April 2018 by using the strategy [“video games” OR “exergaming”] in free text with no limit applied to the search strategies. 

The Medline search through PubMed resulted in 4923 primary studies, 12,802 in ISI Web of Knowledge, and 15,655 in SCOPUS.

A first selection of relevant studies was made after the elimination of duplicate references (*N* = 233) and the exclusion of articles based on the title or abstract that did not mention the use of active video games (*N* = 33,211 references) (see Figure 1).

A **total of** 263 studies were selected for full reading and were reviewed in duplicate. Figure 1 shows the flowchart for identifying primary studies and reporting on the reasons for exclusion. Inclusion and exclusion criteria are detailed in Table 1. A total of 248 references were excluded after reading the full text and 15 of the references met the inclusion criteria.

We also checked the reference lists of the resulting articles found, as well as the references included in meta-analysis and systematic reviews identified through the search. Finally, we attempted to search for unpublished primary studies through manual and electronic searches. For the identification of those ongoing studies, we searched the electronic database of clinical trial registries: Current Controlled Trials, National Health Service—The National Research Register and Clinical Trials. One article [42] was added after this checking resulting in 16 finally included studies.

One person completed full abstraction (CHJ), and a second person verified extractions (MS). Data were checked again before analysis.

### 2.4. Assessment of Methodological Quality

The methodological quality assessment of in each primary included study was carried out by using the ‘Quality Assessment Tool for Quantitative Studies (EPHPP)’ [43], designed by the effective Public Health Practice Project [43].

The questionnaire includes eight dimensions or components:(1)Selection bias;(2)Design;(3)Confounders;(4)Blinding;(5)Data collection methods;(6)Withdrawals and dropouts;(7)Intervention integrity; and(8)Analysis.

Each dimension contains one or more questions or items. The questionnaire is accompanied by a dictionary as a manual for the assessment of the completion of each item and for the interpretation of the final assessment in each dimension. This dictionary is available [44].

For the first six dimensions, three different types of scores are given: “weak”, “moderate”, and “strong”, where “strong” is the category for higher quality corresponding to lower probability of bias. Finally, the questionnaire allows an overall assessment with the same categories: “weak”, “moderate”, and “strong”. The overall rating is considered “strong” if none of the dimensions scored has the lowest rating of “weak”; it is considered “moderate” if only one of the dimensions has the lowest rating of “weak”; and “weak” if two or more of the dimensions have the lowest rating of “weak”.

In the questionnaire, each qualitative category of each dimensions is given a score: “weak (one point)”, “moderate (two points)”, and “strong (three points)”. This allowed to add the scores of the six dimensions. Recoding “weak” as zero points, “moderate” as one point and “strong” as two points, therefore, gives an overall rating range of 0–12 points in the controlled studies and 0–10 points in the non-controlled studies (pre + post studies with only the intervention group).

Quality assessment followed the recommendations of Chalmers [45] and Santibañez [46] in order to minimize observer bias. Each primary study was assessed independently by two reviewers (CHJ and MS). In those cases of discrepancy in the evaluation, it was assessed whether the discrepancy affected the qualitative rate or the quantitative score, resolving by consensus. Only three discrepancies between the reviewers occurred. In the case of the assessment of the studies by Maddison et al. (2011) [37] and Christison et al. (2016), the discrepancies between the two reviewers in the dimension “withdrawals and dropouts” affected the score, so that it was agreed to use the numerical mean between the two quantitative scores and the qualitative rate for each rate was reported. In the case of the last discrepancy regarding Trost et al. study (2013) study in the dimension “confounding”, it did not affect the score or the qualitative rate of the dimension.

### 2.5. Data Analyses

We chose the standardized mean of difference (SMD) with its 95%CI as a summary measure of effect to allow us to combine data for BMI Kg/m^2^, Z score, or percentile in a single meta-analysis. If a study reported both outcomes, we used the non-standardized data as the first option. The z-score BMI and the percentile BMI were used as second and third option, respectively. This strategy, which is consistent with the approach taken in other reviews [47,48,49], increases the pool of studies, thereby increasing the power to detect a difference in weight change intra-group and between-groups.

In a second strategy, as a sensitivity analysis, the results were grouped for each BMI type: crude (Kg/m^2^), z-score and percentile. In this approach the mean difference (MDs) on the natural scale (not standardized) was used.

When studies provided more than one determination (for example, one at shorter follow up and one at longer follow-up), in a first approach all the determinations were meta-analyzed and, subsequently, determinations were restricted to shorter and longer follow up, respectively.

To weight intervention effects, a random-effects versus fixed effect model was chosen after studying the heterogeneity for each outcome. Statistical heterogeneity was assessed through the Cochran’s Q-test and I^2^ statistics, which describe the percentage of total variation across studies that is attributable to statistical heterogeneity rather than to chance. I^2^ values of 25, 50, and 75% correspond to low, moderate, and high between-study statistical heterogeneity. A *p* value < 0.10 was set as the cut-off point for a statistically significant heterogeneity in the χ^2^ test for heterogeneity [50]. We used the DerSimonian and Laird random effects model with inverse variance to generate SMDs and mean differences (MDs) [51].

A priori established subgroup analyses were performed by: study design (controlled vs. non-controlled), date of publication (≤2010 vs. >2010), place of study (Europe, USA, Australia and New Zealand), design of controlled studies (randomized vs. non-randomized), time at determination of post BMI (weeks of follow up <12 weeks, ≥12–<24 weeks, ≥24 weeks), basal BMI percentile (≥85 percentile vs. BMI was not an inclusion criterion), minimum age of children or adolescents as an inclusion criterion (<12 years vs. ≥12), gender (male vs. both sexes or unspecified sex), and methodological quality score (quantitative score <8 vs. ≥8 points).

Cohen’s (1988) criteria [52] were followed to assess effect size (<0.2 = very small effect; ≥0.2 to <0.5 = small effect; ≥0.5 to <0.8 = medium effect; ≥0.8 = large effect).

We sought evidence of publication bias using the funnel plot method and Egger’s regression asymmetry test [53,54]. In addition, Duval and Tweedie´s “trim and fill” approach was used to obtain the best estimation of the unbiased effect size in mortality [55].

The meta-analysis was written following the recommendations of the Preferred Reporting Items for Systematic Reviews and Meta-Analyses (PRISMA) statement [56]. All analyses were conducted by using Comprehensive Meta-Analysis (CMA v2) [57].

## 3. Results

According to selection criteria, 16 original articles were identified. Table 2 presents a summary of the main methodological aspects of these identified studies, including design, characteristics of patients and interventions, duration, and outcomes analyzed. Ten studies were randomized controlled clinical trials [21,31,37,38,40,58,59,60,61,62]. One study was a non-randomized controlled clinical trial [63]. All of the 11 controlled studies above were two parallel branches (arms) clinical trials. The remaining five trials were non-controlled clinical trials (only one intervention arm) [36,39,42,64,65].

The overall results of the quality analysis for the primary studies finally included are presented in Table 3.

There was variability in the overall quality scores of the different studies, with a range between 1 (Fernández-Rosado, 2013) [65] and 10 (Staiano, 2017) [61] between the lowest and highest scoring studies. Eleven studies [21,31,36,38,39,42,59,60,63,64,65] scored below 8, and five studies [37,40,58,61,62] scored greater than or equal to 8 points.

In the eleven controlled clinical trials, the overall mean quantitative score according to the dictionary was 7.1 out of a total of 12 points. The five non-controlled studies obtained a mean quality score of 3.6 out of 10 points.

The lack of information in Fernandez-Rosado’s study [65] due to the fact that only the abstract from a congress was published, made it the study with the lowest score.

### 3.1. Intra-Group Pre-Post Difference in the BMI in the Intervention Group

Among the 16 included studies, Foley et al (2014) study reported only data for the between-group assessment, so 15 out of the 16 studies provided pre-post intra-group data enough in relation to the intervention with active video games susceptible to be meta-analyzed [21,31,36,37,38,39,40,42,58,60,61,62,63,64,65]. Four of these 15 studies provided two determinations (one at shorter follow up and one at longer follow-up) [31,37,42,62]. Thus, in this first approach, 19 determinations were meta-analyzed.

Figure 2 shows the SMDs in a basic subgroup analysis depending on whether the study was non-controlled (one arm) or controlled (two arms). The individual studies presented variable results, mostly showing a positive effect of the active video games (negative DEMs), but without reaching statistical significance.

The individual results of the 19 determinations from the 15 studies that fulfilled the inclusion criteria, showed a high heterogeneity among them (Q = 105.33, df = 18, *p* < 0.001, I^2^ = 82.91%, τ = 0.50) (see Table 4). The overall intra-group effect of the intervention based on active video games was in favor of the intervention, reaching statistical significance using the fixed effects model: SMD = −0.138; 95%CI (−0.237 to −0.038), *p* = 0.007 and was of borderline statistical significance in the random effects model: SMD = −0.191; 95%CI (−0.386 to 0.003), *p* = 0.053.

Determinations were also restricted to the longer and shorter respectively for each article. By restricting the determinations to the longer follow-up, the overall intra-group effect size was slightly larger: SMD = −0.190; 95%CI (−0.409 to 0.029), *p* = 0.089 than the shorter follow up: SMD = −0.138; 95%CI (−0.331 to 0.054), *p* = 0.159 under the random effects model (see Figure 3; Figure 4).

### 3.2. Intra-Group Pre-Post Difference in the BMI in the Control Group

Figure 5 shows the SMD in the control group for the 10 controlled clinical trials. Three of these studies [31,37,62] provided two determinations (one at shorter follow up and one at longer follow-up). Thus, in this approach, 13 determinations were meta-analyzed in the control group. The overall intra-group effect in the control group was close to zero (SMD= −0.039 and 0.087 under the fixed and random model, respectively), presenting the individual results also high heterogeneity among them (Q = 59.45, df = 12, *p* < 0.001, I^2^ = 79.81%, τ = 0.44) (see Table 5).

By restricting the determinations to the longer follow-up and shorter follow up, the overall intra-group SMD was −0.007 and 0.061 respectively under the fixed model, and 0.112 and 0.127 under the random effects model (see Figure 6 and Figure 7).

### 3.3. Between-Groups Differences in the BMI.

Eight studies provided between group data regarding the non-standardized BMI (Kgr/m^2^) [21,31,37,38,58,59,60,63]. Two of these eight studies provided two determinations (one at shorter follow up and one at longer follow-up) [31,37]. Thus, in this approach, 10 determinations were meta-analyzed.

Figure 8 shows the MDs in the non-standardized BMI (Kgr/m^2^), after making this restriction to controlled studies. The meta-analyzed MD for the non-standardized BMI (Kg/m^2^) between the intervention and control group was in favor of active video games, yielding statistical significance: MD = −0.317, 95% CI (−0.442 to −0.193), *p* < 0.001 in the random effects model. The individual results of the 10 determinations from the 8 studies, showed a low and moderate heterogeneity among them (Q = 12.40, df = 9, *p* = 0.191, I^2^ = 27.44%, τ = 0.10). See Table 6.

By restricting the determinations to the longer and shorter follow-up, the meta-analyzed MD were −0.285; 95%CI (−0.472 to −0.097), *p* = 0.003 and −0.365; 95%CI (−0.504 to −0.227), *p* < 0.001, respectively, under the random effects model (see Figure 9 and Figure 10).

Five studies provided between group data regarding the BMI z-score [37,40,59,61,62]. Two studies provided two determinations [37,62] so in this approach seven determinations were meta-analyzed. Figure 11 shows the MDs as regards to BMI z-score. The meta-analyzed MD with respect to the z-score BMI was also in favor of active video games and statistically significant: MD = −0.077, 95%CI (−0.139 to −0.016), *p* = 0.013 (random effects models), presenting the individual results a higher heterogeneity among them (Q = 257.18, df = 7, *p* < 0.001, I^2^ = 97.28%, τ = 0.09) (see Table 7).

By restricting the determinations to the longer and shorter follow-up, the meta-analyzed MD for the z-score BMI (random effects models) were −0.057 and −0.069, respectively (see Figure 12 and Figure 13). 

### 3.4. Publication Bias

Regarding the publication bias in relation to the active video games intra-group BMI difference (all determinations and all follow-ups), the Funnel plot visually presented an asymmetry with a greater number of studies to the right (against active video games). When incorporating the Duval and Tweedie “trim and fill” procedure, the model included five studies from the left (represented as black circles). The best adjusted (unbiased) estimate using the Duval and Tweedie´s “trim and fill” procedure was, therefore, more in favor of intervention with an adjusted overall DEM of −0.297 and −0.365 using the fixed-effect and random-effects model respectively for all determinations in all follow-ups (see Figure 14 and Figure 15).

Egger’s test marked the intersection value at −2.130; *p*-value (unilateral contrast test) = 0.063; *p*-value (bilateral contrast test) = 0.127.

Regarding BMI difference between groups, the funnel plot also visually presented an asymmetry with a larger number of studies to the right (against active video games), with three studies included on the left when incorporating the Duval and Tweedie “trim and fill” procedure in relation to crude BMI (kgr/m^2^). This also supports the interpretation of our results in the difference between groups in a conservative way as well to be possibly slightly underestimated (see Figure 16 and Figure 17).

### 3.5. Subgroup Analysis

When the intervention was applied to children and adolescents with a basal BMI percentile more than or equal to 85 (overweight or obese), a greater intra-group effect was showed in favor of the intervention with a SMD= −0.483, 95%CI (−0.862 to −0.105) *p* = 0.012 under the random effects model (see Figure 18).

Regarding quality, studies that scored higher in the quality assessment showed more favorable intra-group results for active video games. SMD = −0.696.

These results and the rest of a priori established subgroup analyses described in the methods section, can be consulted in the Appendix A (see Appendix A).

## 4. Discussion

In relation to the pre-post intra-group difference concerning the group with active video games (intervention group), the size of the overall effect according to the Cohen criteria (1988) [52] would be small with a SMD close to 0.2 under the random effect model. Four studies reported BMI determinations at more than one time follow-up. By restricting the determinations to the longer follow-up, the overall effect size was slightly larger (SMD = −0.190) than the shorter follow up (SMD = −0.138).

Eleven out of the 15 included studies that provide pre-post intragroup data were controlled clinical trials (two arms) [21,31,37,38,40,58,59,60,61,62,63] and five studies were non-controlled (only one intervention arm) [36,39,42,64,65]. The effect of active video games in non-controlled studies was less without heterogeneity in outcomes.

The effect of active video games in the eleven controlled studies was greater, but with a high heterogeneity among the individual results.

One study stood out for its impact on results and heterogeneity [62]. This study reported the BMI Z-score, in two follow-ups (at eight and 16 weeks). In this study, both in the control group and in the intervention group with active video games, an active exercise program was additionally applied. This would explain their different results, more in favor than in the rest of the studies.

The overall effect in the pre-post intra-group difference concerning the control group was close to zero (SMD close to zero). When an intervention was given in the control group [21,31,62], an increase in the effect size was observed in favor of non-active video games intervention. As mentioned above, in the Trost et al. study (2013) [62], a physical exercise program was also applied to the control group, explaining its results. In the study by Adamo et al. (2010) [21] the control group exercised on a static bike while listening to music. In the study by Graves et al. (2010) [31] only sedentary video games were used, with no scheduled physical exercise interventions. These two studies showed more homogeneous results close to null.

Only one study [63] evaluated BMI with a long-term follow up (50 weeks). Its results support a between-groups positive difference in favor of active video games. However, the intra-group results of this study were different from those of the other studies, with a small effect size against active video game based intervention (SMD= 0.257) and a slightly greater effect size against the control group (SMD = 0.517). Excluding their results, heterogeneity in the subgroup of studies without any intervention in the control group disappeared, reducing the intra-group SMD to 0.024 in the control group. It should be noted that, in addition to being the only study with long-term follow-up, it was the only non-randomized controlled study. Regarding the difference between groups, their results would, in any case, support a positive difference in favor of active video games. That is, although children and teens who play active video games increase their BMI, they would do so to a lesser extent than children without any intervention. Long-term effects, however, need to be evaluated in a larger number of studies so that their results can be susceptible to meta-analysis.

The study of Fernandez-Rosado et al. (2013) [65] also showed a very favorable result for active video games. This study also presents differential characteristics, since it is the study that scored the worst in the quality analysis due to the fact that only one abstract has been published at a congress, but no article was published according to our knowledge.

Thus, the differentiated results in effect size and heterogeneity provided by the studies by Trost et al. (2013) [62], Azevedo et al. (2014) [63], and Fernandez-Rosado (2013) [65] can be explained by their different design or quality.

Regarding the predefined subgroup analysis, when the active video games intervention was applied to children and adolescents with a basal BMI percentile greater than or equal to 85 (overweight or obese), a greater effect in favor of the active video games was showed. In relation to the date of publication, the first published studies (in 2010 or earlier) showed a greater homogeneity of the results around the null effect. From the year 2010 onwards, the heterogeneity of the results was greater, due to the fact that the three aforementioned studies were published in 2013 and 2014, and also probably linked to the evolution of the active video games with a greater number of them and a higher variability among them, together with the greater heterogeneity in the clinical trial designs themselves.

In relation to the choice of model for meta-analysis, when restricting to non-controlled studies, there was no heterogeneity in the results, supporting the choice of a fixed-effect model. Nevertheless, due to a priori identified sources of heterogeneity, and the overall high heterogeneity for the rest of analysis, we considered it appropriate to show results under both models. 

Regarding the difference between groups, in relation to the comparison of the crude BMI (Kgr/m^2^), the intervention based on active video games obtained a MD of −0.352 or −0.317 depending on the use of a fixed or random effects model based on the determinations provided by eight studies [21,31,37,38,58,59,60,63]. This means that, on average, the intervened group obtained a decrease of 0.3 units more in the crude BMI with respect to the control group. By restricting the determinations to the longer follow-up (MD = −0.285), the size of the overall effect was slightly smaller with respect to the shorter follow up (MD = −0.365). In all cases these differences between groups reached statistical significance. In relation to the comparison of the Z-score BMI between groups, a statistically significant MD of −0.077 units was also obtained based on the determinations provided by five studies [37,40,59,61,62]. Thus, these differences do not seem to be of great clinical relevance, but they are in favor of the intervention, and reach statistical significance.

The fact that active video games represent another alternative for physical practice, together with the positive aspects of adherence because of the motivation and enjoyment that they provide [15,25,66,67], makes them an excellent instrument for health promotion. They also have other advantages, such as their easy use, low cost, and the fact that they can be used in different environments, such as the home itself. One of the limitations of active video games would be their need for space to be used, which is not always the case. They are also not exempt from risks, such as muscular skeletal injuries, and accidents with household objects have being reported [68,69]. It is also possible that, like traditional (non-active) video games, misuse of them can be associated with problems of addiction or increased aggressiveness [70]. Active video games should, therefore, be seen as a complement to an active lifestyle, but not as a substitute for real physical exercise.

Publication bias represents a particular threat to the validity of meta-analysis [71]. Due to our comprehensive search strategy, omission of important published trials seems unlikely. However, as regards the publication bias in the active video games intra-group BMI difference (all determinations and all follow-ups), the funnel plot visually presented an asymmetry with a greater number of studies to the right (against active video games), suggesting the existence of a publication bias. Thus, when incorporating the Duval and Tweedie “trim and fill” procedure, the model included five studies from the left and the best adjusted (unbiased) estimate using the Duval and Tweedie´s “trim and fill” procedure was, therefore, more in favor of intervention.

Egger’s test results marked the intersection value not very close to 0. Their null hypothesis is that there is no publication bias, so the closeness to statistical significance, seen in the context of the lack of power of this test as a limitation, would not rule out publication bias, but would support it [53,54].

Therefore, despite the limitations of graphs (visual subjectivity) and statistical tests (lack of statistical power), it can be assumed that our results may be underestimated by publication bias. Thus, the results of the meta-analysis can be interpreted as conservative, possibly being affected by a publication bias. Unbiased estimation would be more favorable to the use of active video games. In any case, following Cohen’s (1988) criteria for assessing effect size, even in the best unbiased estimate, this would be ≥ 0.2 to < 0.5, corresponding to a small effect size.

Regarding BMI difference between groups, the Funnel plot also visually presented an asymmetry with a larger number of studies to the right (against active video games), with three studies included on the left when incorporating the Duval and Tweedie “trim and fill” procedure. This also supports the interpretation of our results in the difference between groups in a conservative way being possibly slightly underestimated.

Meta-analysis protocols recommend quality assessment of the primary studies included in the review. In our case, the quality of the studies identified was assessed using a blind, standardized, and systematic procedure. Two investigators independently reviewed each of the studies and discrepancies were resolved by consensus. The use of the EPHPP questionnaire for the methodological quality assessment versus other alternatives was decided given the variability in the designs of the studies finally included in the meta-analysis (both uncontrolled and controlled and within these both randomized and non-randomized). The Quality Assessment Tool for Quantitative Studies [43] is designed to evaluate both controlled and uncontrolled studies simultaneously, while other options, such as the Cochrane Handbook for Systematic Reviews of Interventions [54], is designed only for randomized and controlled clinical trials (RCTs).

There was great variability in the quality of the different studies. The risk that they were affected by biases was high or could not be ruled out in most of them. By restricting to higher quality studies, heterogeneity decreased and the positive effect of the intervention was reinforced, reaching the five studies that scored highest in quality, a large effect size in favor of intervention based on active video games (SMD = −0.696).

As a conclusion, our meta-analysis show a statistically significant effect in favor of using active video games on BMI in children and adolescents. Given the relative novelty of active video games, and that the analyzed studies have evaluated short-term interventions, it is necessary to conduct long-term studies with larger sample sizes in order to reach valid and reliable comparisons. The clinical relevance of the positive effect must be also evaluated. On the other hand, intervention with active video games is very heterogeneous between studies, both in terms of devices, type of games, or duration and frequency of use. This heterogeneity is also due to technological innovation in this field over the years. It is necessary to continue investigating the benefits of these video games, trying to standardize the interventions, in order to generalize them in children and adolescents in terms of public health.

## Figures and Tables

**Figure 1 ijerph-16-02424-f001:**
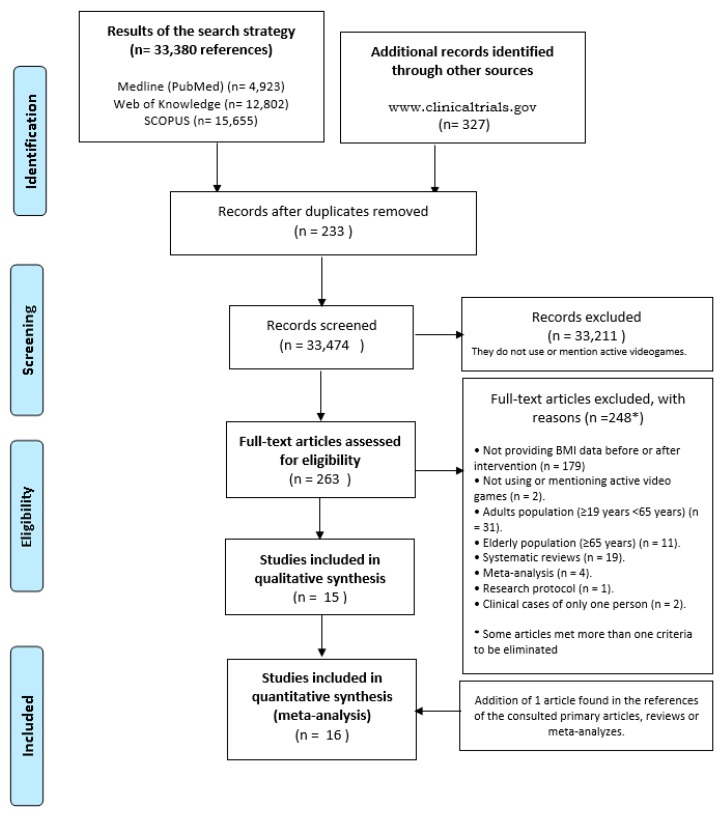
PRISMA 2009 Flow Diagram. Diagram of identification of clinical trials studies whose intervention was based on active video games, and its variable result was measured in BMI (Kg/m^2^, Zscore or percentile). 1 January 2000 to 13 April 2018.

**Figure 2 ijerph-16-02424-f002:**
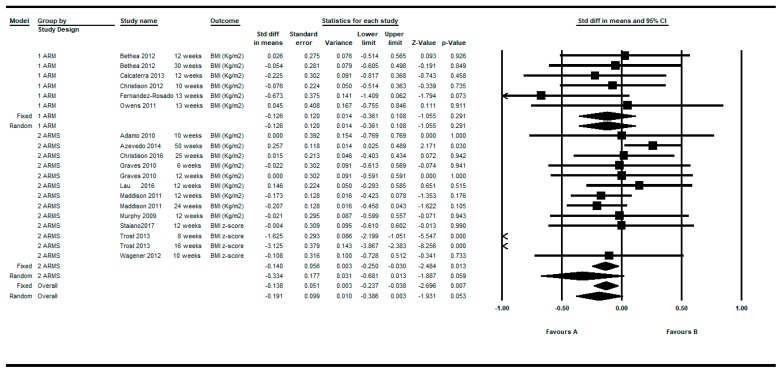
Pre-post intra-group difference in BMI in the group intervened with active video games. All measurements and all follow-ups.

**Figure 3 ijerph-16-02424-f003:**
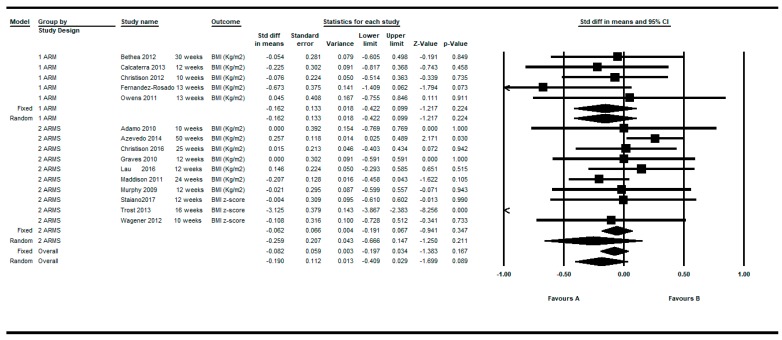
Pre-post intra-group difference in BMI in the group intervened with active video games. All measurements in the longest follow-up.

**Figure 4 ijerph-16-02424-f004:**
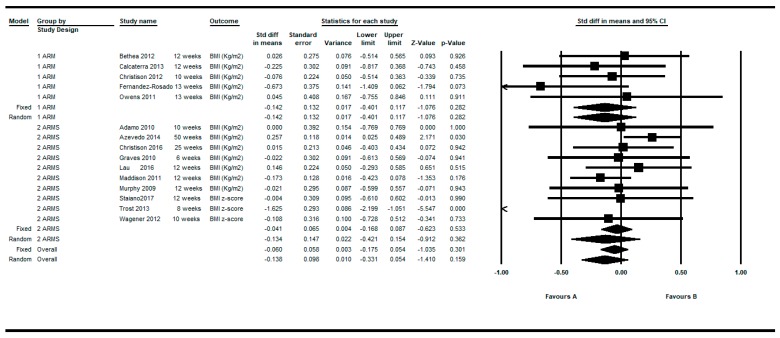
Pre-post intra-group difference in BMI in the group intervened with active video games. All measurements in the shortest follow-up.

**Figure 5 ijerph-16-02424-f005:**
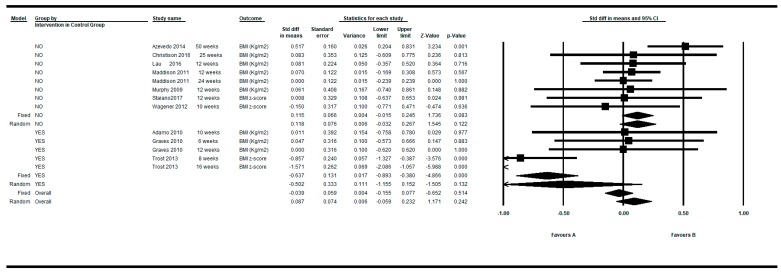
Pre-post intra-group difference in BMI in the control group. All measurements and all follow-ups.

**Figure 6 ijerph-16-02424-f006:**
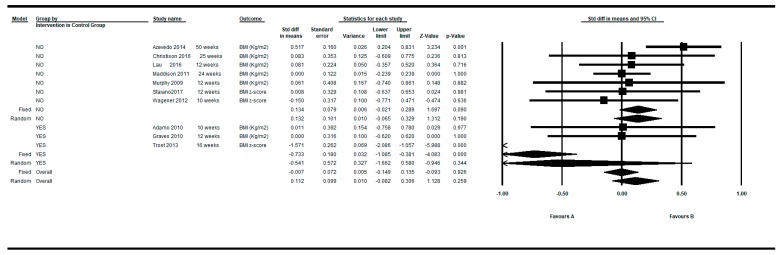
Pre-post intra-group difference in BMI in the control group. All measurements in the longest follow-up.

**Figure 7 ijerph-16-02424-f007:**
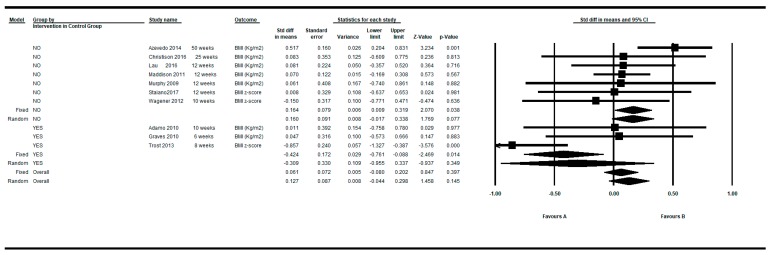
Pre-post intra-group difference in BMI in the control group. All measurements in the shortest follow-up.

**Figure 8 ijerph-16-02424-f008:**
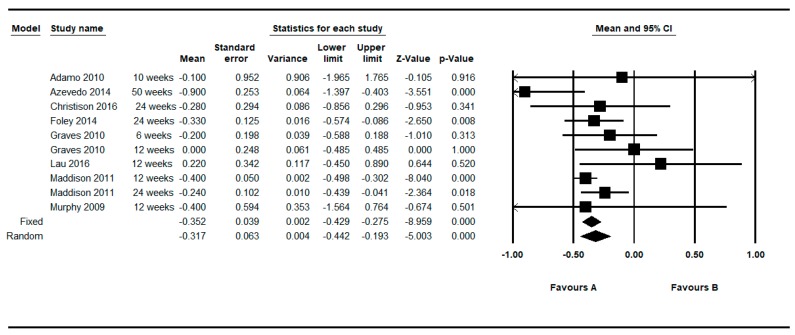
Pre-post difference of BMI between groups. Only BMI (Kgr/m^2^) and all follow-ups.

**Figure 9 ijerph-16-02424-f009:**
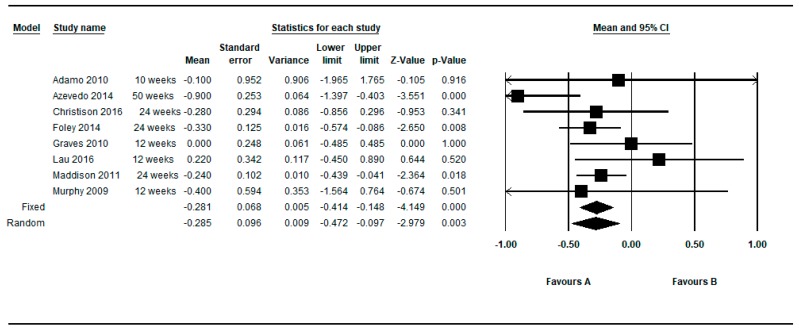
Pre-post difference of BMI between groups. Only BMI (Kgr/m^2^) in the longest follow-up.

**Figure 10 ijerph-16-02424-f010:**
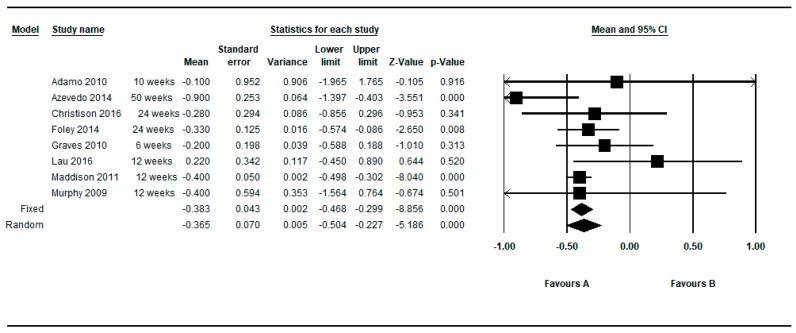
Pre-post difference of BMI between groups. Only BMI (Kgr/m^2^) in the shortest follow-up.

**Figure 11 ijerph-16-02424-f011:**
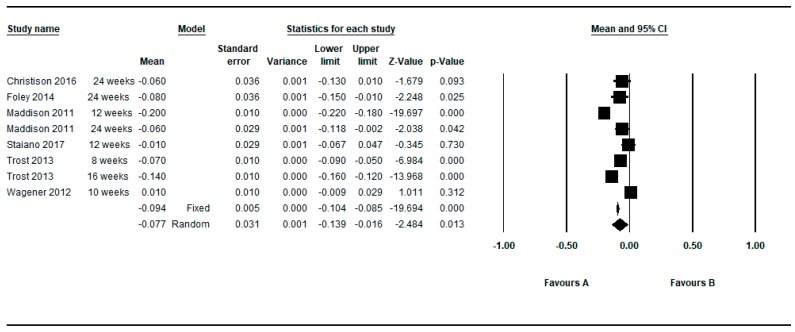
Pre-post difference of BMI between groups. Only BMI z-score and all follow-ups.

**Figure 12 ijerph-16-02424-f012:**
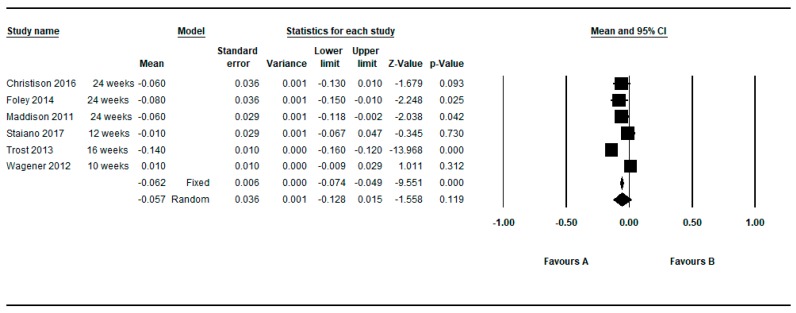
Pre-post difference of BMI between groups. Only BMI z-score in the longest follow-up.

**Figure 13 ijerph-16-02424-f013:**
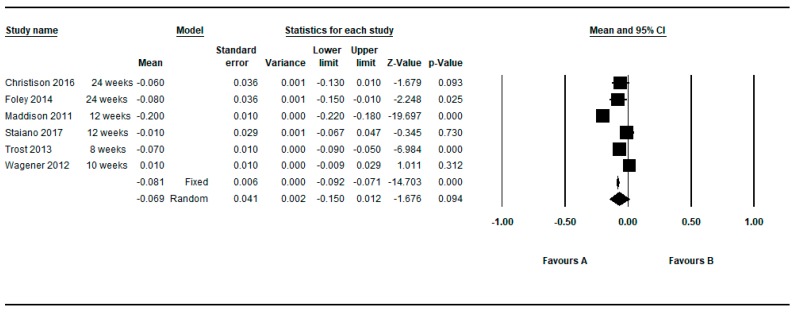
Pre-post difference of BMI between groups. Only BMI z-score in the shortest follow-up.

**Figure 14 ijerph-16-02424-f014:**
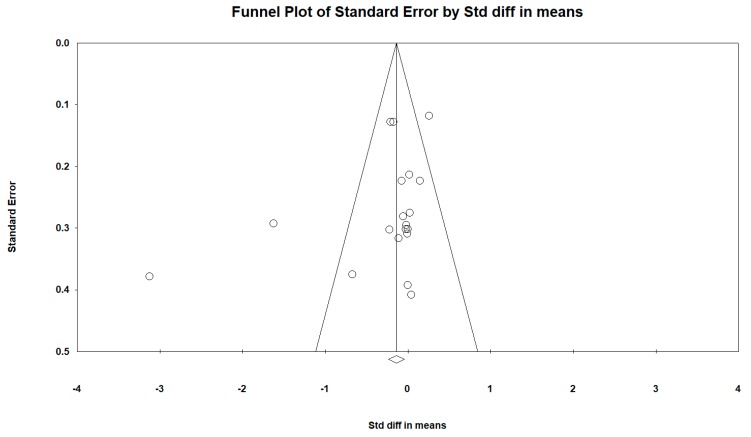
Funnel diagram or “funnel plot” of the standard error (Y axis) by the SMD (X axis). Pre-post intra-group difference in BMI in the group intervened with active video games (all determinations and all follow-ups).

**Figure 15 ijerph-16-02424-f015:**
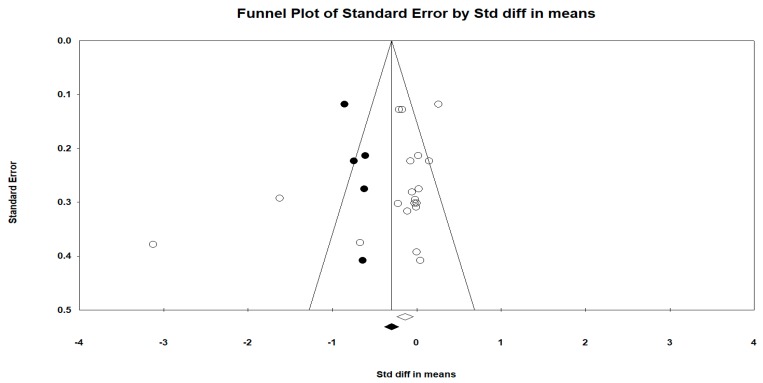
Funnel diagram or “funnel plot” of the standard error (axis of the Y) by the SMD (axis of the X), after incorporating the Duval and Tweedie “trim and fill” procedure. Pre-post intra-group difference in BMI in the group intervened with active video games (all determinations and all follow-ups).

**Figure 16 ijerph-16-02424-f016:**
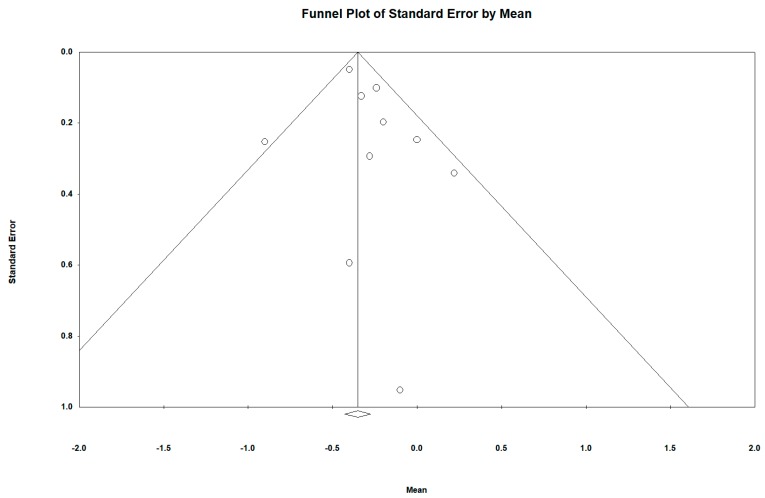
Funnel diagram or ‘funnel plot’ of the standard error (axis of the Y) by the DM (x axis). Pre-post difference of BMI (Kgr/m2) between groups and all follow-ups.

**Figure 17 ijerph-16-02424-f017:**
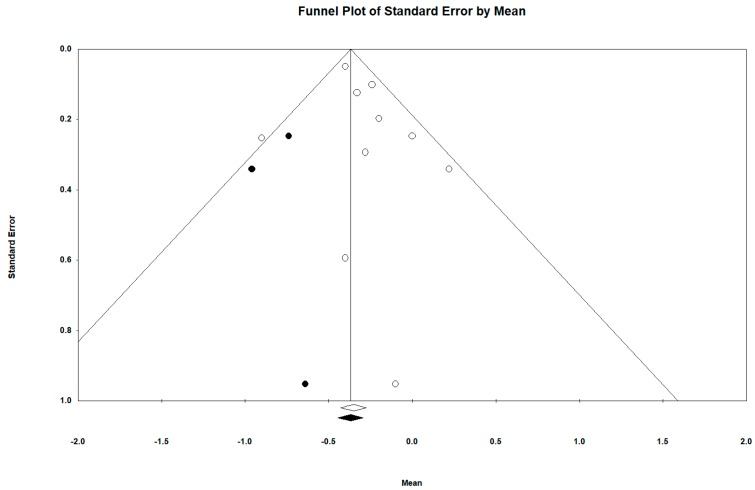
Funnel diagram or “funnel plot” of the standard error (axis of the Y) by the DEM (axis of the X), after incorporating the Duval and Tweedie “trim and fill” procedure. Pre-post difference of BMI (Kgr/m^2^) between groups and all follow-ups.

**Figure 18 ijerph-16-02424-f018:**
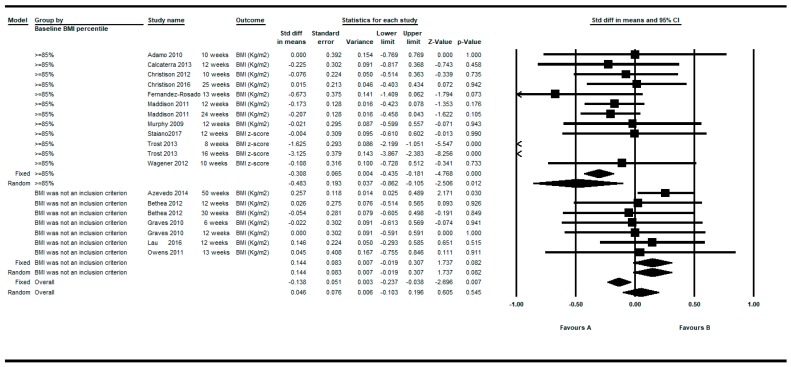
Pre-post intra-group difference in BMI in the group intervened with active video games. Subgroup analysis based on baseline BMI (≥85 vs. BMI was not an inclusion criterion).

**Table 1 ijerph-16-02424-t001:** Inclusion and exclusion criteria.

**Inclusion Criteria**
• Children and adolescents (≤18 years old). Design: clinical trials both controlled (randomized or quasi-randomized) and non-controlled. • Intervention based on active video games. • Language: written in English or Spanish. • Information on at least one determination of the BMI (Kg/m^2^, Z score, or percentile) both pre and post intervention, or differences between groups in the BMI in case of controlled studies.
**Exclusion Criteria**
• Adults (≥19 years <65 years) or elderly (≥65 years). • Non-epidemiological studies: clinical cases of only one person.

**Table 2 ijerph-16-02424-t002:** Characteristics of included studies.

First Author. Publication Year.	Country	Study Design	Study Population	Duration (Weeks)	Procedure	Main Results	Other Results
Adamo et al., 2010 [21]	Canada	Randomized controlled trial.	26 participants. 12–17 years. Inclusion criterion: BMI ≥ 85th percentile with at least 1 metabolic complication; or BMI ≥ 95th percentile.	10 weeks	Intervention group (*n* = 13): They used active videogame (GameBike). Control group (*n* = 13): They used stationary cycling while listening to music. Both: Sessions 60 minutes 2 times per week during 10 weeks.	BMI (kg/m^2^): Intervention group: Pre: 35.5 ± 9.3 Post: 35.5 ± 9.7 Control group: Pre: 39.9 ± 8.9 Post: 39.4 ± 8.9 BMI percentile: Intervention group: Pre: 97.8 ± 1.4 Post: 97.5 ± 1.8 Control group: Pre: 97.8 ± 2.7 Post: 97.8 ± 2.3	Heart rate. Intervention group: Pre: 177.1 ± 21.4 Post: 169.1 ± 19.2 Control group: Pre: 172.2 ± 16 Post: 163.2 ± 23.1
Azevedo et al., 2014 [63]	UK	Non-randomized controlled trial.	189 participants. 11–13 years. BMI was not an inclusion criterion: Average baseline BMI intervention group = 20.4; DE [4.2]. Average baseline BMI control group = 19.5; DE [3.3].	50 weeks	Intervention group (*n* = 117). They used dancing active videogame (dance mat). Control group (*n* = 72). Non-intervention.	BMI (kg/m^2^): Intervention group: Pre: 20.4 ± 4.2 Post: 21.5 ± 4.4 Control group: Pre: 19.5 ± 3.3 Post: 21.3 ± 3.7	
Bethea et al., 2012 [42]	USA	Non-controlled randomized trial.	28 participants. 9–11 years. Only men. BMI was not an inclusion criterion: Average baseline BMI group = 19.8; DE [3.9]. Average baseline BMI percentile = 68.4; DE [28.7].	12–30 weeks	Intervention group (*n* = 28): They used dancing active videogame DDR (Dance Dance Revolution) 3 days per week about 30 min.	BMI (kg/m^2^): Intervention group: Pre: 19.8 ± 3.9 Post (12 weeks): 19.9 ± 3.9 Post (30 weeks): 19.6 ± 3.5 BMI z-score: Intervention group: Pre: 0.71 ± 1.05 Post (12 weeks): 0.66 ± 1.1 Post (30 weeks): 0,55 ± 1 BMI percentile: Intervention group: Pre: 68.4 ± 28.7 Post (12 weeks): 67.1 ± 30.6 Post (30 weeks): 64.5 ± 28.8	Heart rate. Intervention group: Pre: 86 ± 11 Post: 86 ± 15
Calcaterra et al., 2013 [64]	Italy	Non-controlled randomized trial	22 participants. 9–16 years. Inclusion criterion: BMI ≥ 95th percentile.	12 weeks	Intervention group (*n* = 22): Training program for sedentary obese children, including active video games.	BMI (kg/m^2^): Intervention group: Pre: 32.9 ± 4.3 Post: 31.9 ± 4.6 BMI z-score: Intervention group: Pre: 2.52 ± 0.55 Post: 2.37 ± 0.67	VO_2_ max (maximal oxygen consumption) Intervention group: Pre: 47.5 ± 6.5 Post: 51.3 ± 6.3
Christison et al., 2012 [36]	USA	Non- controlled randomized trial	40 participants. 8–16 years. Inclusion criterion: BMI ≥ 85th percentile.	10 weeks	Intervention group: The Exergaming for Health program was designed to be multidisciplinary and of moderate intensity. The curriculum had 3 main components: facilitated activity with exergaming, nutrition education, and behavioral management discussions. Subjects participated in 10 weekly 1hour facilitated activity sessions. There were five one-hour exergaming sessions in the first half of the program and five one-hour combined exergaming/traditional exercise sessions during the latter portion of the program.	BMI (kg/m^2^): Intervention group: Pre: 31.07 ± 6.41 Post: 30.59 ± 6.25 BMI z-score: Intervention group: Pre: 2.24 ± 0.41 Post: 2.17 ± 0.49	
Christison et al., 2016 [58]	USA	Randomized controlled trial.	48 participants 8–12 years. Inclusion criterion: Overweight or obese	25 weeks	Intervention group (*n* = 35): Two main components: facilitated group activity with active videogaming and family-centered didactics focused on improved nutrition education and behavioral aspects of lifestyle change. Control group (*n* = 13): Non-intervention	BMI (kg/m^2^): Intervention group: Pre: 28.8 ± 6.4 Post (25 weeks): 28.9 ± 6.6 Control group: Pre: 29.5 ± 4.8 Post (25 weeks): 29.9 ± 4.8 BMI z-score: Intervention group: Pre: 2.13 ± 0.4 Post (25 weeks): 2.07 ± 0.5 Control group: Pre: 2.25 ± 0.3 Post (25 weeks): 2.25 ± 0.3	Heart rate. Intervention group: Pre: 87 ± 12 Post: 77 ± 10 Control group: Pre: 87.2 ± 8.7 Post: 81.2 ± 9.9
Fernández Rosado. 2013 [65]	Spain	Non-controlled randomized trial	15 participants 7–13 years. Inclusion criterion: Obese	13 weeks	Intervention group (*n* = 15): They provided 60 minutes PlayStation Eye Toy USB camera maximum day.	BMI (kg/m^2^): Intervention group: Pre: 27.91 ± 2.55 Post: 25.84 ± 3.52	
Foley et al., 2014 [59]	New Zealand	Randomized controlled trial.	322 participants. 10–14 years. Inclusion criterion: Overweight or obese.	24 weeks	Intervention group (*n* = 160): They used active videogame (PlayStation Eye Toy USB). Control group (*n* = 162): Non-intervention	BMI (kg/m^2^): Intervention group: Pre: 25.64 ± 4.08 Control group: Pre: 25.75 ± 4.25 BMI z-score: Intervention group: Pre: 1.26 ± 1.11 Control group: Pre: 1.25 ± 1.1	
Graves et al., 2010a [31]	UK	Randomized controlled trial.	42 participants 8–10 years. BMI was not an inclusion criterion: Average baseline BMI intervention group = 18.9; DE [4.5]. Average baseline BMI control group= 19.7; DE [4.3].	6–12 weeks	Intervention group (*n* = 22): They used active videogame. Control group (*n* = 20): They used traditional sedentary videogame.	BMI (kg/m^2^): Intervention group: Pre: 18.9 ± 4.5 Post (6 weeks): 18.8 ± 4.5 Post (12 weeks): 18.9 ± 4.6 Control group: Pre: 19.7 ± 4.3 Post (6 weeks): 19.9 ± 4.3 Post (12 weeks): 19.7 ± 4.2	
Lau et al., 2016 [60]	China	Randomized controlled trial.	80 participants 8–11 years.	12 weeks	Intervention group (*n* = 40): They played an active videogame, Xbox 360, twice per week during after-school hours, each for 60 min over 12 weeks in duration. Control group (*n*= 40): Non-intervention	BMI (kg/m^2^): Intervention group: Pre: 19.41 ± 3.63 Post (12 weeks): 19.95 ± 3.78 Control group: Pre: 19.75 ± 3.61 Post (12 weeks): 20.04 ± 3.51	VO_2_ max (maximal oxygen consumption) Intervention group: Pre: 42.93 ± 1.47 Post: 45.21 ± 2.15 Control group: Pre: 42.62 ± 1.75 Post:43.38 ± 2.36
Maddison et al., 2011 [37]	New Zealand	Randomized controlled trial.	258 participants 10–14 years. Inclusion criterion: Overweight.	12–24 weeks	Intervention group (*n* = 123): They used active videogame. (PlayStation Eye Toy USB). Control group (*n* = 135): Non-intervention	BMI (kg/m^2^): Intervention group: Pre: 25.6 ± 4.1 Post (12 weeks): 24.9 ± 4 Post (24 weeks): 24.8 ± 3.6 Control group: Pre: 25.8 ± 4.3 Post (12 weeks): 25.5 ± 4.3 Post (24 weeks): 25.8 ± 4.2 BMI z-score: Intervention group: Pre: 1.3 ± 1.1 Post (12 weeks): 1.1 ± 1.2 Post (24 weeks): 1.1 ± 1.1 Control group: Pre: 1.3 ± 1.1 Post (12 weeks): 1.3 ± 1.1 Post (24 weeks): 1.3 ± 1	
Murphy et al., 2009 [38]	USA	Randomized controlled trial.	35 participants 7–12 years. Inclusion criterion: BMI ≥ 85th percentile.	12 weeks	Intervention group (*n* = 23): They used active videogame. DDR (Dance Dance Revolution), 1 session per week. Control group (*n* = 12): Non-intervention	BMI (kg/m^2^): Intervention group: Pre: 27.9 ± 4.8 Post: 27.8 ± 4.7 Control group: Pre: 31.8 ± 5 Post: 32.1 ± 4.9	VO_2_ max (maximal oxygen consumption) Intervention group: Pre: 27.1 ± 5.4 Post: 29.5 ± 4.5 Control group: Pre: 25.6 ± 5.4 Post: 24.3 ± 4.8
Owens et al., 2011 [39]	USA	Non-controlled randomized trial	20 participants 12 children (8–13 years.) BMI was not an inclusion criterion: Average baseline BMI group = 19.4; DE [4.4].	13 weeks	Intervention group: They used active videogame Wii FitTM during 13 weeks daily.	BMI (kg/m^2^): Intervention group: Pre: 19.4 ± 4.4 Post: 19.6 ± 4.4	VO_2_ max (maximal oxygen consumption) Intervention group: Pre: 34.3 ± 9.6 Post: 38.4 ± 8.6
Staiano et al., 2017 [61]	USA	Randomized controlled trial.	38 participants Girls. 14–18 years. Inclusion criterion: Overweight or obese.	12 weeks	Intervention group (*n* = 20): They practiced three sessions of 1 h dance active video per week for 12 weeks. Control group (*n* = 18): Non-intervention	BMI z-score: Intervention group: Pre: 2.1 ± 0.5 Post: 2.098 ± 0.5 Control group: Pre: 2.1 ± 0.5 Post: 2.104 ± 0.501	
Trost et al., 2013 [62]	USA	Randomized controlled trial.	69 participants 8–12 years. Inclusion criterion: BMI ≥ 85th percentile.	8–16 weeks	Intervention group (*n* = 31): They used two types of active video games during 16 weeks. Control group (*n* = 38): They performed a program of exercise without active video game.	BMI z-score: Intervention group: Pre: 2.14 ± 0.08 Post (8 weeks): 2.01 ± 0.08 Post (16 weeks): 1.89 ± 0.08 Control group: Pre: 2.16 ± 0.07 Post (8 weeks): 2.1 ± 0.07 Post (16 weeks): 2.05 ± 0.07	
Wagener et al., 2012 [40]	USA	Randomized controlled trial.	40 participants. 12–18 years. Inclusion criterion: BMI ≥ 95th percentile.	10 weeks	Intervention group (*n* = 20): They used dancing active videogame. (dance pads) 3 times per week 40 min the first session, and 75 min for following sessions. Control group (*n* = 20): Non-intervention	BMI z-score: Intervention group: Pre: 3.15 ± 0.19 Post: 3.13 ± 0.18 Control group: Pre: 3.15 ± 0.2 Post: 3.12 ± 0.2	

**Table 3 ijerph-16-02424-t003:** Results of the quality assessment based on the ‘Quality Assessment Tool for Quantitative studies (EPHPP)’, for the 16 studies finally included.

	Selection Bias (0–2 Points)	Study Design (0–2 Points)	Confounders (0–2 Points)	Blinding (0–2 Points)	Data Collection Methods (0–2 Points)	Withdrawals and Dropouts (0–2 Points)	Final Decision Based on the Dictionary	Final Mark (RCT and CCT 0–12 Points) (Pre + Post 0–10 Points)
Adamo 2010	W (0)	S (2)	W (0)	W (0)	S (2)	S (2)	W	6
Azevedo 2014	W (0)	S (2)	S (2)	W (0)	S (2)	W (0)	W	6
Bethea 2012	W (0)	M (1)	NA	W (0)	S (2)	S (2)	W	5
Calcaterra 2012	W (0)	M (1)	NA	W (0)	S (2)	W (0)	W	3
Christison 2012	M (1)	M (1)	NA	W (0)	W (0)	S (2)	W	4
Christison 2016	S (2)	S (2)	M (1)	M (1)	S (2)	M/W (0.5) *	S/M *	8.5
Fernández-Rosado 2013	W (0)	M (1)	NA	W (0)	W (0)	W (0)	W	1
Foley 2014	M (1)	S (2)	S (2)	W (0)	W (0)	W (0)	W	5
Graves 2010	W (0)	S (2)	S (2)	W (0)	S (2)	M (1)	W	7
Lau 2016	M (1)	S (2)	W (0)	W (0)	S (2)	S (2)	W	7
Maddison 2011	M (1)	S (2)	S (2)	W (0)	S (2)	S/M (1.5) *	M	8.5
Murphy, 2009	M (1)	S (2)	S (2)	W (0)	W (0)	W (0)	W	5
Owens 2011	W (0)	M (1)	NA	W (0)	S (2)	S (2)	W	5
Staiano 2017	M (1)	S (2)	S (2)	M (1)	S (2)	S (2)	S	10
Trost 2013	W (0)	S (2)	S (2) *	W (0)	S (2)	S (2)	W	8
Wagener 2012	W (0)	S (2)	S (2)	M (1)	M (1)	S (2)	M	8

RCT = Randomized controlled trial, CCT = Controlled clinical trial (not randomized). Pre + Post = Non-controlled trial. W = “weak”, M = “moderate” and S = “strong”. NA = Not applicable. * In the case of the assessment of the studies by Maddison et al. (2011), and Christison et al. (2016), the discrepancies between the two reviewers in the dimension “withdrawals and dropouts” affected the score, so that it was scored with the average mean of the scores (1.5 and 0.5 points respectively). In the case of the discrepancy regarding Trost et al. study (2013) in the dimension “Confounding”, it did not affect the score or the qualitative rate of the dimension.

**Table 4 ijerph-16-02424-t004:** Global heterogeneity in the group intervened with active video games, and depending on their design (1 or 2 arms). All measurements and all follow-ups.

Intervention Group	N of Determinations	Heterogeneity					
		Q	df	p (χ^2^)	I^2^ (%)	τ^2^	τ
**All determinations (15 studies, 4 with two determinations)**	19	105.33	18.00	0.00	82.91	0.25	0.50
1 arm	6	2.83	5.00	0.72	0.00	0.00	0.00
2 arms	13	102.49	12.00	0.00	88.29	0.33	0.58

**Table 5 ijerph-16-02424-t005:** Global heterogeneity in the control group, and depending on whether or not in this control group any intervention not based on active video games was practiced. All measurements and all follow-ups.

Control Group	N of Determinations	Heterogeneity					
		Q	df	p (χ ^2^)	I^2^ (%)	τ^2^	τ
**All determinations (10 studies, three with two determinations)**							
All studies	13	59.45	12.00	0.00	79.81	0.19	0.44
No intervention in the control group	8	8.20	7.00	0.31	14.71	0.007	0.08
Any intervention not based on active video games	5	24.98	4.00	0.00	83.98	0.46	0.67

**Table 6 ijerph-16-02424-t006:** Global heterogeneity between groups. Only BMI (Kgr/m^2^) and all follow-ups.

Difference between Groups. BMI (Kgr/m^2^).	N of Determinations	Heterogeneity					
		Q	df	p (χ^2^)	I^2^ (%)	τ^2^	τ
**All determinations (eight studies, two with two determinations)**	10	12.40	9.00	0.19	27.44	0.01	0.10

**Table 7 ijerph-16-02424-t007:** Global heterogeneity between groups. Only BMI z-score and all follow-ups.

Difference between Groups. IMC z-Score.	N of Studies	Heterogeneity					
		Q	df	p (χ^2^)	I^2^ (%)	τ^2^	τ
**All determinations (6 studies, 2 with two determinations)**	8	257.18	7.00	0.00	97.28	0.01	0.09

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
