# Peer review of "Impact of Active Video Games on Body Mass Index in Children and Adolescents: Systematic Review and Meta-Analysis Evaluating the Quality of Primary Studies"

_ijerph, 2019, doi:10.3390/ijerph16132424_

Round 1
Reviewer 1 Report
Excellent paper, well written, valid methods and sound results and discussion.
This meta analysis was very thoroughly done, the statistics are accordingly and the paper is well written. The paper is concise and exactly given the results and discussion, not too short and not
too long!
I advise to include also the Funnel plot to illustrate the publication bias.
Author Response
Response:
The Funnel plots to illustrate the publication bias were included as supplementary Material (S fig 1-4). Now they are included as new Figures 14-17.
Reviewer 2 Report
This is a good piece of work on systematic review and meta-analysis including design and methodology. I do have a few clarifications as follows:
The definition of the active video games had not been identified. How did the active video games correlated with the reduction of BMI?
Were those interventional studies?
What are the roles of public health and environmental interventions on prevention and reduction childhood obesity based on review with active video games?
Why set the age range under 18 years old?
Author Response
Response:
Active video games are those that allow players to interact physically, through their body movements (arms, legs or the whole body), allowing the physical interaction of players and their movements with the virtual reality that appears on screen through different devices. We have explained a little more extensively what are the active video games in the introduction (lines 45-48).
Evidence on the effects of active videogames on the promotion of physical activity, energy expenditure, oxygen volume consumption and heart rate is based on interventional studies, that could be one-arm trials (with pre-post determinations in the intervention group), or two-arm controlled trials, incorporating a control group, to determine differences between the two groups.
Our hypothesis was that this higher energy expenditure is also associated with a decrease in Body mass index, so active video games could be a public health and environmental intervention on prevention and reduction of childhood obesity. The fact that active videogames represent another alternative for physical practice, together with the positive aspects of adherence because of the motivation and enjoyment that they provide in children and adolescents, makes them an instrument for health promotion. Nevertheless, active videogames should be seen as a complement to an active lifestyle, but not as a substitute for real physical exercise.
We set the age range under 18 years old, because the upper limit of age according to included primary studies was 18 years in Wagener et al., 2012 or Staiano et al., 2017 (see table 2, study population column). The upper limit of age range from 10 to 16 years in the rest of included studies. All the included study population can be considered as children or adolescents.
Reviewer 3 Report
Interesting manuscript with good potential to add to the literature on active gaming and BMI in children/youth. It would be useful to have more details on the team who conducted the analysis within the methods section of the paper, for example, how many reviewers were involved? It is not until the discussion that it is noted that two reviewers were involved and the EPHPP Quality Assessment Tool was used. What process was used for confirming among the research team? Were there any disagreements or discrepancy among the two reviewers? If yes, how was agreement obtained? Also, it would be useful to have more of a critique of the process and whether the EPHPP was a useful tool. Only cursory mention is given to why it was used in the discussion section. Finally, what are the recommendations for further research in this area. More elaboration on this could be provided. For example, what would represent "high quality" studies to continue to advance this line of inquiry? What are questions that remain after the analysis? What should be further explored moving forward?
Author Response
Response:
In the methods section, lines 133-135, it was specified that “Data collection followed the recommendations of Chalmers [45] and Santibañez [46] in order to minimize observer bias: each article was evaluated independently by two reviewers (CH and MS). In cases of disagreement, the final evaluation was obtained by a consensus meeting between them”.
We have changed the paragraph indicating more clearly that it refers also to the evaluation of the Quality of the articles, and explaining in greater depth number of discrepancies and the methodology followed in the cases of discrepancies
Discrepancies were only observed on three occasions in the valuations of overall items.
In the Trost et al. study (2013) a reviewer assessed that there were important differences between groups prior to the intervention, and confounding bias was possible. The other reviewer based the assessment only on the statistical significance, without considering these nos statistically significant differences as important enough to cause confounding bias. This discrepancy did not affect the qualitative classification of dimension 3 (confounding), because the confounders (the variables with differences) were actually controlled either in the design or analysis, so the dimension was rated as S “Strong” for the two reviewers.
In the study by Maddison et al. (2011) the withdrawals and dropouts are different in Figure 1 and Table 2 of the article (24 weeks). One reviewer based on Figure 1 and the other on Table 2, with the qualitative classification of dimension 6 (withdrawals and dropouts) being different: S “Strong” for one reviewer and M “Moderate” for the other reviewer. Nevertheless, this did not, affect the overall qualitative rating of the article considerered as M “Moderate”.
Regarding quantitative assessment, it was agreed to use the numerical mean between the two quantitative scores (1.5 points).
The same occurred in the study by Christison et al. (2016). The discrepancies in the dimension "withdrawals and dropouts" affected the quantitative score, so that it was scored with the average mean of the scores (0.5 points).
In relation to last paragraph of the “2.5. Assessment of methodological quality” section, now it reads: “Quality assessment followed the recommendations of Chalmers [45] and Santibañez [46] in order to minimize observer bias. Each primary study was assessed independently by two reviewers (CH and MS). In those cases of discrepancy in the evaluation, it was assessed whether the discrepancy affected the qualitative rate or the quantitative score, resolving by consensus. Only three discrepancies between the reviewers occurred. In the case of the assessment of the studies by Maddison et al. (2011), and Christison et al. (2016), the discrepancies between the two reviewers in the dimension "withdrawals and dropouts" affected the score, so that it was agreed to use the numerical mean between the two quantitative scores and the qualitative rate for each rate was reported. In the case of the last discrepancy regarding Trost et al. study (2013) in the dimension "Confounding", it did not affect the score or the qualitative rate of the dimension”.
All this information is showed also in table 3 results and table 3 footnotes
* In the case of the assessment of the studies by Maddison et al. (2011), and Christison et al. (2016), the discrepancies in the dimension "withdrawals and dropouts" affected the score, so that it was scored with the average mean of the scores (1.5 and 0.5 points respectively). In the case of the discrepancy regarding Trost et al. study (2013) in the dimension "Confounding", it did not affect the score or the qualitative rate of the dimension.
The ‘Quality Assessment Tool For Quantitative studies (EPHPP)’ presents limitations, but as it is explained in the discussion section, EPHPP was the unique approach up to our knwodledge designed to evaluate both controlled and uncontrolled studies simultaneously, while other options that we have used such as the Cochrane Handbook for Systematic Reviews of Interventions is designed only for randomised and controlled clinical trials (RCTs). In our study we have five non controlled trials among the 16 original articles identified according to selection criteria.
Among the limitations of the questionnaire our study context, they stood out:
Dimension 1: Selection bias. With respect to the mode of selection of the study population, in no study was it found to be done by random sampling or by a procedure that would ensure that the sample was most likely representative of the target population (item 1). However, the percentage of participation was high in some studies (item 2). Based on the dictionary, both conditions are necessary to obtain a "strong" rating, so no study could be classified as "strong".
Dimension 2: Sudy design. In this dimension the dictionary points with the same score of "strong" all randomized studies, both those that describe the randomization method, and therefore can be assessed if it is appropriate or not, and those that do not specify the randomization method. Even a controlled but non-randomized study scores on the basis of the dictionary as "strong". One-arm (Pre+Post) studies, although not controlled, all score as "moderate".
Dimension 4: Blinding. The masking of the intervention (active videogames) is not possible. However, It is possible that the response variables have been measured blindly (without knowing intervention status or not). In only one study (Wagener et al., 2012) was the collection of response variables blind. Based on the dictionary, the evaluation of this study was "moderate", and of "weak"in the rest.
According to these limitations, as sensitivity analysis It was decided to take a complementary quantitative approach by reassigning scores in these dimensions, and also adding a score for the last two dimensions (7 and 8) that do not score in the dictionary. The results of metaanalysis according to subgroups by quality did not change and did not change either the interpretation of results and conclusions.
Response:
Given the relative novelty of the phenomenon, the research is at an early stage, and the studies found have evaluated short-term interventions. Long-term studies with larger sample numbers are needed focusing also in the clinical relevance of the effect. On the other hand, intervention with active video games is very heterogeneous between studies, both in terms of devices, type of games or duration and frequency of use. This heterogeneity is also due to technological innovation in this field over the years. It is necessary to continue investigating the benefits of these videogames, trying to standardize the interventions, in order to generalize them in children and adolescents in terms of public health.
We have added this last paragraph in the discussion section.